# Administration of lower doses of radium-224 to ankylosing spondylitis patients results in no evidence of significant overall detriment

**Nicholas D. Priest[1], Lawrence T. Dauer[2], David G. Hoel[3]\***

**1** Radiobiology and Health, Canadian Nuclear Laboratories, Chalk River, Ontario, Canada, **2** Department of Medical Physics, Memorial Sloan-Kettering Cancer Center, New York, New York, United States of America, **3** Medical University of South Carolina, Charleston, South Carolina, United States of America

\* dghoel@gmail.com

**Data Availability Statement:** The European Radiobiology Archive website can be accessed from: https://www.bfs.de/EN/bfs/science-research/

## Abstract

The use of low doses of radium-224 ($^{224}$Ra) chloride for the treatment of ankylosing spondylitis was stopped following the discovery that patients treated with it had a higher than control incidence of leukaemia and other cancers. This was so even though the treatment resulted in decreased pain and increased mobility–both of which are associated with decreased mortality. It was decided to re-analyze the epidemiological data looking at all causes of death. The risk of leukaemia, solid cancer, death from non-cancer causes and from all causes in a study populations of men that received either the typical dose of 5.6 to 11.1 MBq of $^{224}$Ra, any dose of $^{224}$Ra or no radium were compared using the Cox proportional hazard model. For patients that received the typical dose of $^{224}$Ra agreed with the excess cancer was similar to that reported in previous studies. In contrast, these patients were less likely to die from non-cancer diseases and from all causes of death than the control patients. No excess mortality was also found in the population of all males that received the radionuclide. It is concluded that $^{224}$Ra treatment administered at low doses to patients with ankylosing spondylitis did not impact mortality from all causes. The study demonstrates the need to consider all causes of death and longevity when assessing health impacts following irradiation.

## Introduction

Ankylosing spondylitis is a chronic, progressive inflammatory disease of the axial skeleton [1]. The disease is associated with severe pain, decreased mobility and increased mortality [2]. Ankylosing spondylitis is treated palliatively to minimise pain and maximize mobility and from early in the twentieth century [3,4] until recently, some patients (in Germany in the 1910's and then again since the 1940's, in France since 1922 and in the UK since 1946 [5]) were treated by the intravenous injection of radium-224 ($^{224}$Ra, alpha-emitting radionuclide, T½ = 3.66 d) chloride solutions. Initially, the amount of $^{224}$Ra administered was very large and two deaths from acute radiation syndrome in 1912 resulted in the

projects/era/era_node.html The ERA contact person is Mandy Birschwilks (mbirschwilks@bfs.de) who provide the data on 21.07.2015. Data are restricted because of the following: Data are owned by a third party (Helmholtz-Center Munich) and were transferred to ERA under the requirement that any design for an additional evaluation has to be approved before forwarding the data, and data contain potentially sensitive information. The data recipient must not forward the data to any other group or person.

**Funding:** The authors received no specific funding for this work.

**Competing interests:** The authors have declared that no competing interests exist.

temporary cessation of its use in Germany. Later in Germany smaller amounts (up to 140 MBq), often in combination with platinum and eosin, as in the drug Peteosthor, were administered [6] but these were also found to be associated with increased mortality–particularly from bone cancers.[7,8]. In addition to bone cancers, Speiss [9] showed that high dose treatments resulted in increases in a wide range of other health effects including: leukaemia, bone fractures, growth retardation in children, cataracts, breast cancer, kidney insufficiency and cardio-vascular disease. It follows that the administration of high doses of $^{224}$Ra were stopped.

More recently, smaller doses were administered (typically 10 MBq) that delivered average skeletal dose of ~0.5 Gy and at these doses no excess osteosarcoma has been seen. Low dose treatments were started at Münster University Orthopaedic Hospital and continued to 1990 when treatments were stopped [10].

However, despite the absence of osteosarcoma, the results of a prospective cohort study, conducted by Wick et al. of 1588 patients treated for ankylosing spondylitis with $^{224}$Ra between 1945 and 1975 demonstrated an excess risk of other malignancies [11]. The highest measured relative risks, compared with matched Saarland (Germany) registry data were for leukaemia (acute myeloid leukaemia, SIR = 3.9; chronic myeloid leukaemia, SIR = 3.0; acute lymphatic leukaemia, SIR = 2.6; all leukaemia, SIR = 2.8). However, a control population of ankylosing spondylitis patients that received no radium treatment also demonstrated more leukaemia than expected based on Saarland data (acute myeloid leukaemia, SIR = 1.5; chronic myeloid leukaemia, SIR = 1.1; acute lymphatic leukaemia, SIR = 2.0; all leukaemia SIR = 1.6) and given the small numbers of leukaemia found (19 in the treated population cf. 12 in the unexposed control population) it was difficult to come to any firm conclusions about the excesses seen for any sub-type. Other excess risks were indicated, including for thyroid cancer, kidney cancer and female genital organ tumours [11,12]. Because of the results of this study and the availability of alternative treatments, the German Federal Institute for Drugs and Medical Devices declared $^{224}$Ra treatments obsolete and revoked the licence for the $^{224}$Ra containing drug SpondylAT. The German Society of Rheumatology also deleted the treatment from its list of recommended treatments.

The withdrawal of $^{224}$Ra as a treatment was mandated even though a significant body of evidence had been accumulated over the years which showed that the treatment provided effective pain relief and facilitated greater mobility within the treated patient population [13–15]. It might be expected that these benefits would have been reflected in decreased mortality associated either with less prolonged drug use or with improved mobility. This possibility was not investigated at that time as only studies that were concerned with cancer mortality were evaluated. Accordingly, it was decided to undertake an additional study that analysed all mortality within the studied $^{224}$Ra-treated and untreated ankylosing spondylitis patient populations.

## Methods

The data used by Wick et al. [10] was archived in the European Radiobiology Archive following the completion of studies. The authors requested permission to use this raw data for the present study. Subsequently, permission was granted and the data was transferred by the Registry to Atomic Energy of Canada Ltd. (now Canadian Nuclear Laboratories) and the Medical University of South Carolina for analysis. Data was provided for 1588 ankylosing spondylitis patients that had received $^{224}$Ra treatment (typically 10 weekly injections–each comprising 1MBq of $^{224}$Ra chloride solution) and 1456 control patients that were treated without the use of radium. For each subject information on patient gender and age, date of entry into the

study, date of treatment, amount of $^{224}$Ra administered, date of death and cause of death (by ICD-07 code) was provided.

As a first step in the analysis, for simplicity, data for the smaller number of female patients were removed from the study. The females consisted of less than 10% of the patients (9.5% exposed, 8.2% controls) and also they exhibit different disease and survival profiles. For the study two populations were selected. The first population comprised all male subjects that had received radium. For the second population, male patients that received unrecorded quantities of radium and those that received either higher (>299µCi (11.1MBq), 410 subjects, mean dose 575µCi (21.3MBq), 26 subjects >1mCi (37.1 MBq), maximum 4.4mCi (163.3MBq)) or lower (<150µCi (5.6MBq), 232 subjects, mean dose 144µCi (5.3MBq), minimum 14µCi (0.5MBq)) than typical radium doses were removed from the analysis (Fig 1).

The analysis of two patient groups was undertaken because those that did not receive the standard treatment dosage (150µCi (5.6MBq) to 299µCi (11.1MBq)) and higher / lower than standard intakes of $^{224}$Ra might be expected to result in atypical health outcomes. The causes of death and survival for both populations of radium-treated patients were compared with those recorded for the 1351 male control ankylosing spondylitis patients that received no radium. The average age at entry into the study of the controls was 43.8y and of the treated patients 37.5y. The average amount of radium given to the second, censored group of treated patients was 274µCi (10.1MBq) which resulted in a calculated average skeletal dose of 0.4Gy and a committed effective dose of 2.9Sv. These doses are similar to those calculated by Lassmann et al. [13] and were calculated using the ICRP biokinetic model for radium as updated in 1993 [16]. Radium retention post-injection in each skeletal compartment was integrated and average tissue doses were calculated assuming the decay of $^{224}$Ra and its progeny within the skeleton. The committed effective dose was calculated using the ICRP models implemented

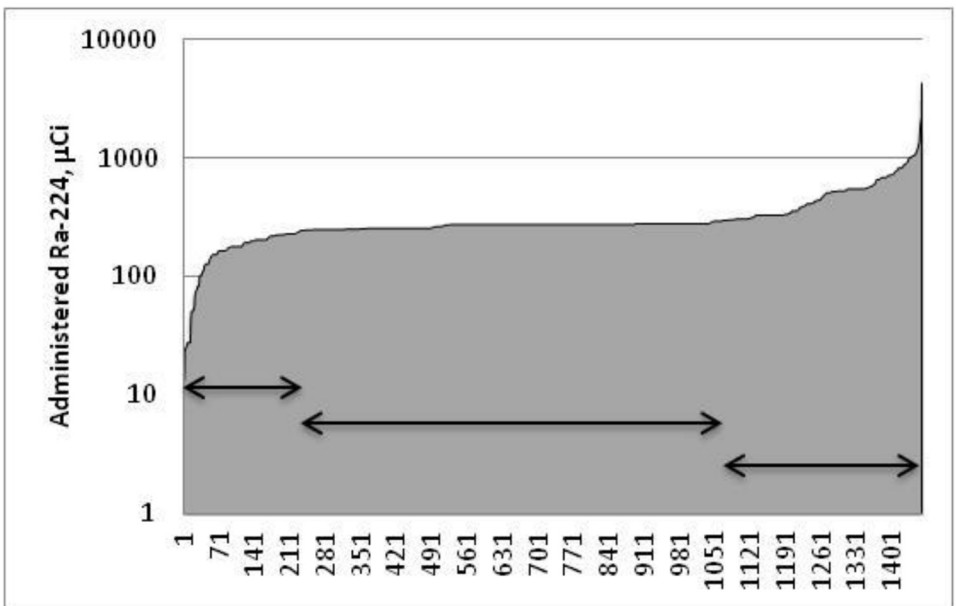

**Fig 1. Graph showing amount of $^{224}$Ra administered by patient number.** 820 male patients (centre arrow) that received the target treatment dose of approximately 10.3MBq of radium chloride were selected for comparison with the control patients.

within the dosimetry code: GenmodPC v5.0 (Canadian Nuclear Laboratories, Chalk River, Canada). The second analysis compared all male patients with the male control population.

The populations were compared using the Cox proportional hazard model. The relative risk of leukaemia, solid cancer, non-cancer death and all causes of death were calculated. For the analysis, it was assumed that the average age at treatment for controls was the archived age, less 0.8y, which corresponded to the difference between 38.7y and 39.5y seen in the treated patient group. Also, it was assumed that there were no differences between the treated and control patients except for the administered $^{244}$Ra. The Cox model allowed for the different ages of the patients entering the study with attained age used as the time scale with age at entry as a covariable. The patients were followed until the age of the end point of interest or the censoring age of a competing risk or age at last follow-up. As described in Wicks et al. [10, 11], the follow-up data was obtained by sending questionnaires to the patients obtained from the hospital records as well as current hospital patient record information. The risk type used in the Cox Model was relative risk. The results for the group of males that received the typical dose of radium were also presented as Kaplan-Meier cumulative risk functions.

In addition, the numbers and percentage of all the males that received radium, of the group of males that received the typical dose of radium and of the group of control males that died from diseases were calculated and grouped according to ICD7 codes. The ICD7 groups employed were: ICD 001 to 138 (infections); ICD 140 to 239 (neoplasms); ICD 240 to 245 (allergic diseases); ICD 250 to 289 (endocrine diseases); ICD 290 to 299 (blood disease); ICD 300 to 326 (psychiatric disease); ICD 330 to 334 (vascular CNS lesions); ICD 400 to 402 (rheumatic fever); ICD 410 to 468 (cardiovascular disease); ICD 470 to 527 (respiratory disease); ICD 530 to 587 (GI tract disease); ICD 590 to 617 (urinogenital disease); ICD 690 to 716 (skin disease); ICD 720 to 749 (musculoskeletal disease); ICD 750 to 776 (congenital abnormality); ICD 780 to 789 (organ /system symptoms); ICD 795 (ill defined /unknown); other–not classified. The results are presented in histogram form.

## Results

The numbers of deaths recorded in the database were 1137 of 1351 (84%) for control males, 484 of 820 (59%) in the selected radium-treated group (cf. 1037 of 1462 (71%) for all treated males). The results of the analysis of relative risk for the smaller, censored male cohort that received (150μCi (5.6 MBq) to 299μCi (11.1 MBq)) of radium and for all male patients are shown in Table 1 and cumulative hazard results are presented in Figs 2–5. Whereas increased

**Table 1. The calculated hazard ratios, p values and 95% confidence intervals for cancer and non-cancer deaths in the male radium patients injected $^{224}$Ra compared to controls.**

|  | Leukaemia Death | Solid Cancer Death | Non-Cancer Death | All Causes of Death |
|---|---|---|---|---|
| Typical radium dose males vs. male controls | | | | |
| Hazard Ratio | 2.56 | 1.15 | 0.81 | 0.88 |
| Probability | 0.08 | 0.26 | <0.001 | 0.012 |
| Lower CI | 0.89 | 0.90 | 0.72 | 0.79 |
| Upper CI | 7.54 | 1.45 | 0.91 | 0.98 |
| All males that received radium vs. male controls | | | | |
| Hazard Ratio | 3.19 | 1.13 | 0.98 | 1.01 |
| Probability | 0.015 | 0.23 | 0.62 | 0.74 |
| Lower CI | 1.26 | 0.93 | 0.89 | 0.93 |
| Upper CI | 8.09 | 1.37 | 1.07 | 1.11 |

## Leukemia Deaths

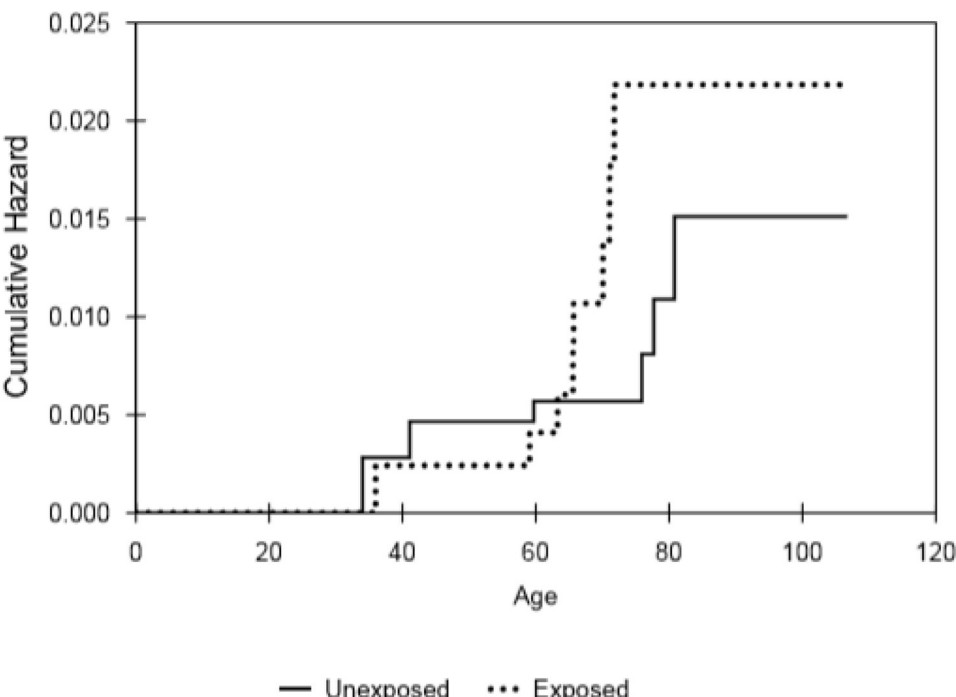

**Fig 2. Cumulative hazard for all leukaemia types, within the selected populations of males that either received the typical dose or did not receive ²²⁴Ra treatment, as a function of attained age.**

deaths from all leukaemia and all solid cancers are indicated, with hazard ratios (HR) approaching significance, (p = 0.08 and p = 0.26, respectively) the relative risk of death from non-cancer was markedly reduced in the male patients treated with radium compared to controls (HR = 0.81; 95% CI = 0.72–0.91; p = <0.001). Similarly, the relative risk of death from all causes was reduced in the selected group of radium patients (HR = 0.88; 95% CI = 0.79–0.98; p = 0.012) that received the typical dose of radium. Table 1 also shows the results for all males that received radium compered with the controls. The hazard ratio for leukaemia (HR = 3.19; 95% CI = 1.26–8.09; p = 0.015) showed a significant excess. However, no significant excesses were found for solid cancers and for non-cancer diseases (HR = 1.13 and 0.98, respectively). Also, there was no overall increase in the risk of death from all causes (HR = 1.01; 95% CI = 0.93–1.11; p = 0.74).

For the relative risk of leukaemia death, by leukaemia type, the results produced (not shown) were like those published by Wick et al. [11] with about twice as many myeloid as lymphatic leukaemias. As previously reported by Wick et al. [11] no osteosarcoma (an expected result of higher radium intakes [7] was found in the radium-treated patients.

The cumulative hazard curves for leukaemia (Fig 2), solid cancer (Fig 3), non-cancer (Fig 4) and all causes of death (Fig 5) produced for the controls and subjects that received the typical dose of radium are presented. These show that there was little difference in mortality within the censored radium-treated and control group up to the age of 60 in the cases of leukaemia deaths, non-cancer deaths and all causes of death. Increases in solid cancer deaths above the levels in controls occurred later at 80+ years.

## Solid Cancer Deaths

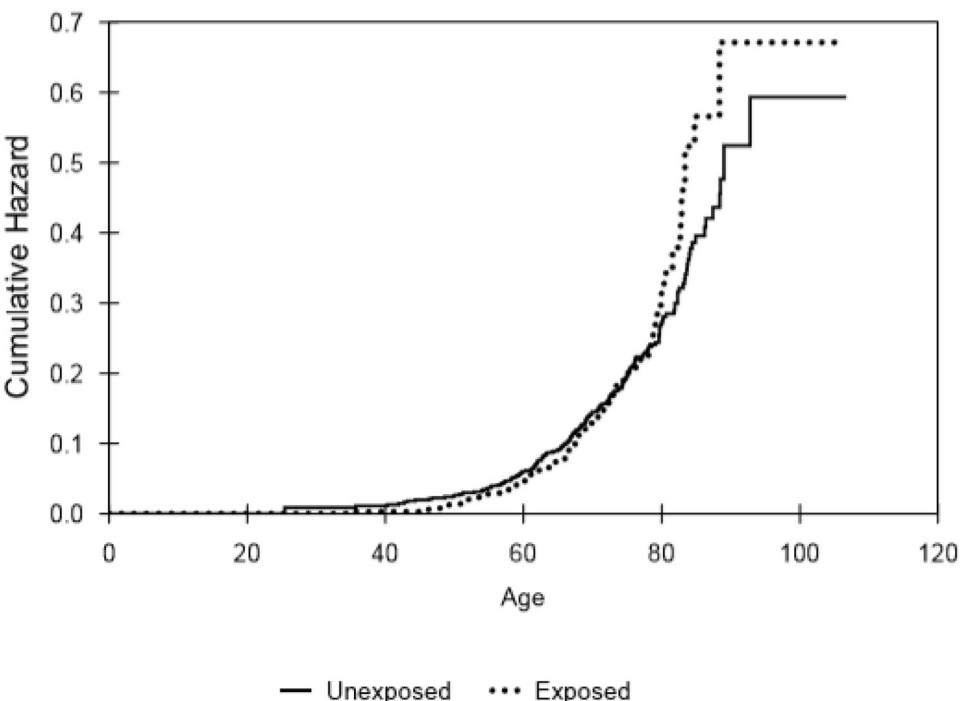

Fig 3. Cumulative hazard for all solid cancer, within the selected populations of males that either received the typical dose or did not receive [224]Ra treatment, as a function of attained age.

Fig 6 presents the percent recorded causes of death by ICD 7 code in the control group and both groups of patients that received [224]Ra. This shows indicated increases in all neoplasms, including leukaemia, (ICD 140 to 239) and in diseases related to organ system failures (ICD 780 to 789) following radium treatment. In contrast, radium treatment resulted in apparent reductions in the percentage of deaths from vascular CNS lesions (ICD 330 to 334), from cardiovascular disease (ICD 410 to 468) and from ill-defined and unknown causes of death (ICD 795). The percentage of deaths from other coded causes were similar in all the groups.

## Discussion

The results showed that [224]Ra treatment at typical levels of 5.6MBq to 11.1MBq in total administered dose had no negative impact on the survival of the ankylosing spondylitis patients compared to the control patients. Indeed, the results showed that the treatments resulted in a significant reduction in the relative risk of death from all causes because of the reduced incidence of non-cancer deaths–most notably in the percentage reduction in deaths from CNS lesions, cardiovascular disease and ill-defined /unknown causes. Moreover, a small increase seen in cancer incidence is to be expected given that cancer and non-cancer are competing causes of death. Similarly, when all males treated with [224]Ra were compared to controls no evidence of excess mortality was seen (HR = 1.01; 95% CI = 0.93–1.11; p = 0.74). It might be expected that in the 410 censored male patients treated with the largest

## Non Cancer Deaths

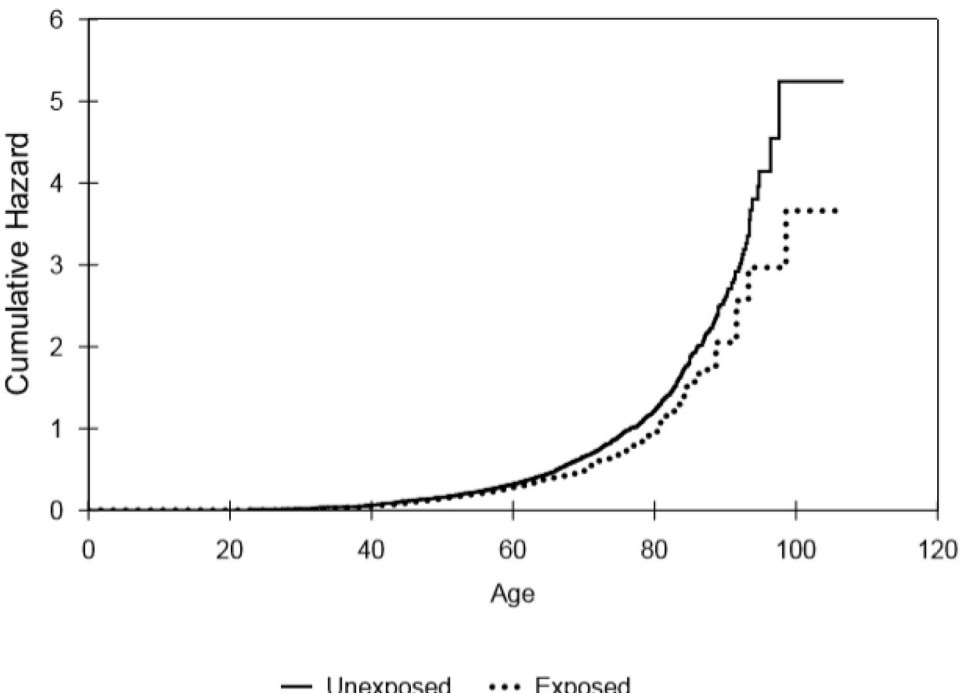

Fig 4. Cumulative hazard for death from non-cancer causes, within the selected populations of males that either received or did not receive the typical dose of $^{224}$Ra, as a function of attained age.

amounts of $^{224}$Ra (resulting in a calculated average skeletal dose of 6.4Gy and a committed effective dose of 46Sv) cancer-deaths may be more important. This was checked by a separate analysis which confirmed a significant excess of non-cancer deaths (HR = 1.17; 95% CI = 1.03–1.34; p = 0.02) in this group of patients. However, the additional $^{224}$Ra administered resulted in no increase in solid cancer frequency (HR = 1.12; 95% CI = 0.84–1.50 cf. RR = 1.14; 95% CI = 0.90–1.44) and no significant increase in leukaemia (HR = 3.01; 95% CI = 0.92–9.88; cf. HR = 2.56; 95% CI = 0.89–7.53) compared to the patients that received the typical dosage of radium. It can be concluded that, at any of the doses employed to treat patients in this cohort of ankylosing spondylitics, no evidence has been produced of increased, radium-induced mortality.

For the calculations it was necessary to assume that the radium-treated and control populations were identical in all respects except for the dose of administered radium. It is noted that a larger percentage of deaths were observed among the controls than the exposed patients. This is because they were on average a little more than 4 years older then exposed patients when entering the study. What is of concern is why some patients attended a hospital that used radiation and others attended a hospital that did not and would it affect the risk analysis results due to possible patient differences. Also, no information on drug treatment was available in the supplied database. Differences, if they existed, in drug use between the radium treated and control patients may, therefore, have influenced the results of the study. For example, it is possible that lower levels of pain-killing NSAIDs were employed in the

## Total Deaths

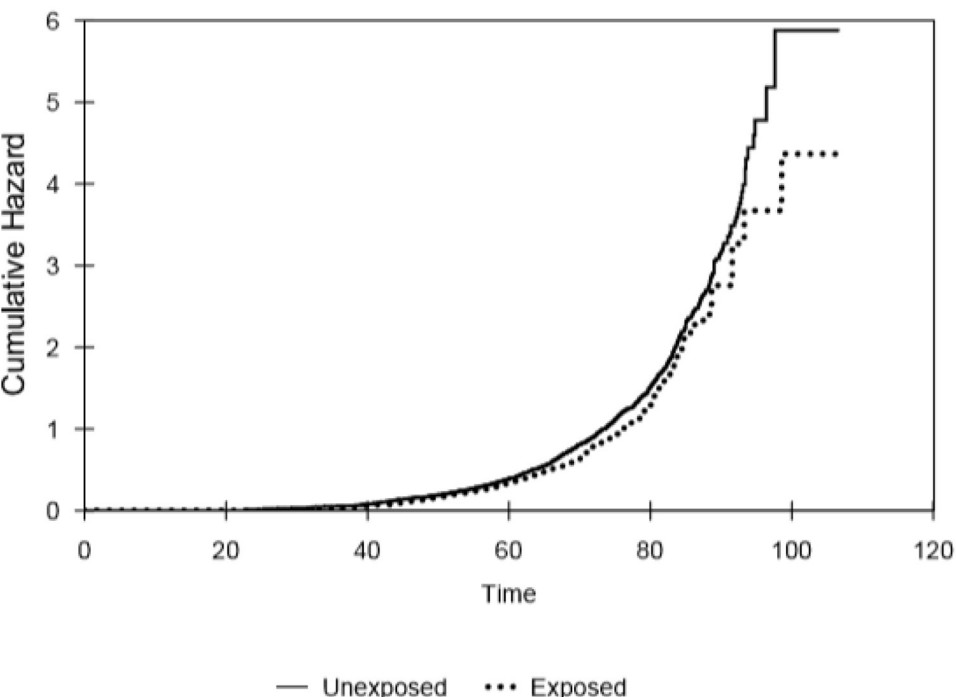

**Fig 5. Cumulative hazard for all causes of death, within the selected populations of males that either received or did not receive typical [224]Ra treatment, as a function of attained age.**

radium-treated patients and this could have influenced the probability of disease. Similarly, increased mobility is reported in [224]Ra-treated patients and this also may have contributed to the lower frequency of non-cancer deaths. Both may have played a role in reducing non-cancer deaths–along with a more general anti-inflammatory effect of the radiation, which could have lowered the prevalence of diseases, such as cardio-vascular disease, which involve inflammation [17]. Inflammation and peritrabecular fibrosis are also associated with osteo-sarcoma [18–20]. Studies with chemical agents hare shown that fibrosis and cancer are con-secutive adverse outcomes that result from the release of DAMPs following the death of cells and the induction of chronic inflammation [21]. Since alpha-particles kill cells it is suggested that these also provide the mechanism for alpha-induced cancer [22]. The absence of insuffi-cient dose (cell death) to induce a state of chronic inflammation could provide an explana-tion why osteosarcoma was not recorded following the administration of low doses of [224]Ra. Osteosarcoma, the typical tumour seen following the administration of high doses of [224]Ra, was not recorded for any of the treated males [7,8].

The results this study add additional information to that reported by Wick et al. [11] who evaluated only cancer mortality data and identified net cancer detriment resulting from the use of [224]Ra. The results of this study indicate the danger of selectively choosing causes of death when trying to assess the overall impact of radiation on mortality. They also demonstrate the need to consider all causes of death and longevity when assessing health impacts following irradiation.

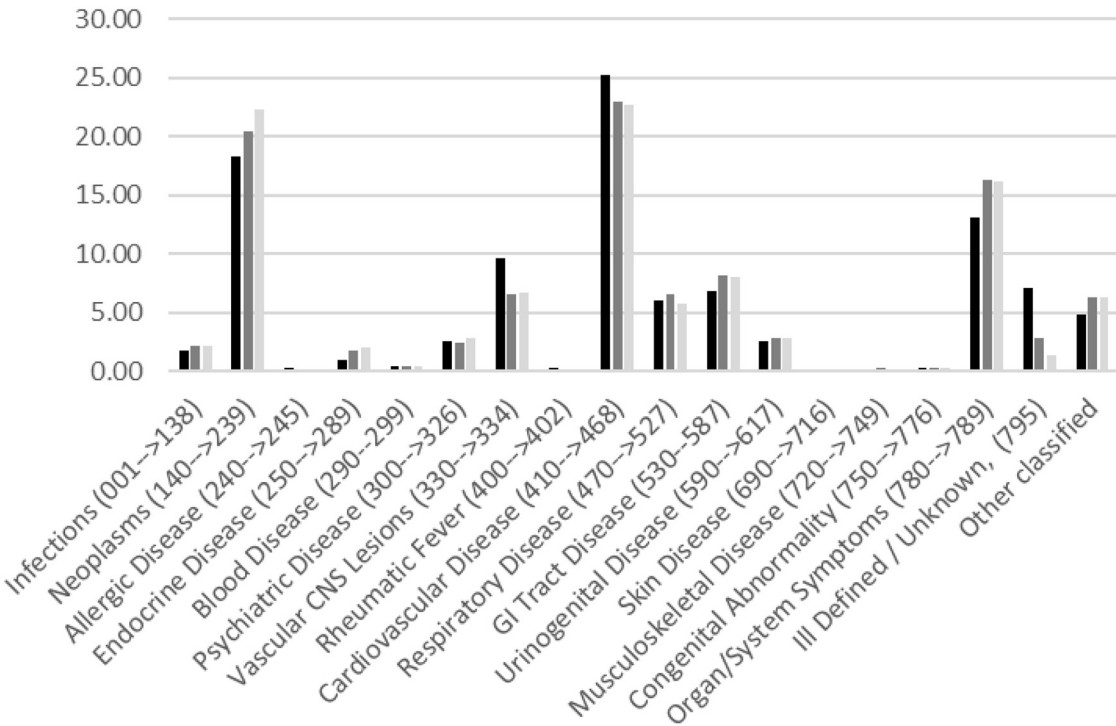

**Fig 6. The percent recorded causes of death in controls (black) all radium-treated males (dark grey) and in patients that received the typical dose of radium (light grey) by ICD 07 code.**

## Author Contributions

**Conceptualization:** Nicholas D. Priest.

**Formal analysis:** David G. Hoel.

**Funding acquisition:** Lawrence T. Dauer.

**Methodology:** David G. Hoel.

**Project administration:** Nicholas D. Priest.

**Software:** David G. Hoel.

**Supervision:** Nicholas D. Priest.

**Writing – original draft:** Nicholas D. Priest.

**Writing – review & editing:** Lawrence T. Dauer.

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
