## [Decision Letter · Decision Letter 0]

26 Nov 2019

PONE-D-19-26797

Administration of Lower Doses of Radium-224 Ankylosing Spondylitis Patients Results in No Evidence of Significant Overall Determent

PLOS ONE

Dear Dr. Hoel,

Thank you for submitting your manuscript to PLOS ONE. After careful consideration, we feel that it has merit but does not fully meet PLOS ONE’s publication criteria as it currently stands. Therefore, we invite you to submit a revised version of the manuscript that addresses the points raised during the review process.

We would appreciate receiving your revised manuscript by Jan 10 2020 11:59PM. To enhance the reproducibility of your results, we recommend that if applicable you deposit your laboratory protocols in protocols.io, where a protocol can be assigned its own identifier (DOI) such that it can be cited independently in the future. For instructions see: http://journals.plos.org/plosone/s/submission-guidelines#loc-laboratory-protocols

We look forward to receiving your revised manuscript.

Kind regards,

Suminori Akiba, M.D., Ph.D.

Academic Editor

PLOS ONE

Journal Requirements:

3. Please include a copy of Table 1 which you refer to in your text.

Additional Editor Comments (if provided):

Reviewers' comments:

Reviewer's Responses to Questions

**Comments to the Author**

1. Is the manuscript technically sound, and do the data support the conclusions?

Reviewer #1: Partly

Reviewer #2: No

2. Has the statistical analysis been performed appropriately and rigorously? 

Reviewer #1: No

Reviewer #2: No

3. Have the authors made all data underlying the findings in their manuscript fully available?

Reviewer #1: Yes

Reviewer #2: Yes

4. Is the manuscript presented in an intelligible fashion and written in standard English?

Reviewer #1: Yes

Reviewer #2: Yes

5. Review Comments to the Author

Reviewer #1: This is a study using the raw data of previous work by Wick RR et. ( Incidence of leukaemia and other malignant diseases following injections of the short-lived alpha-emitter 224Ra into man. Radiat Environ Biophys. 2009 Aug;48(3):287-94) and the authors analysed all mortality within the studied 224Ra-treated and untreated ankylosing spondylitis patient populations. They concluded that 224Ra treatment at typical levels of 5.6 to 11.1MBq in total administered dose had no negative impact on the survival of the ankylosing spondylitis patients compared to the control patients. The results showed that the treatments resulted in a significant reduction in the relative risk of death from all causes because of the reduced incidence of non-cancer deaths – most notably in the percentage reduction in deaths from CNS lesions, cardiovascular disease and ill-defined /unknown causes.

There are some issues requiring further statement.

1. Manuscript style

The manuscript style does not fit the journal's instruction for authors, including the text citation and references writing style. The authors need to revise it.

2. Methods

The authors wrote " As a first step in the analysis, data for the smaller number of female patients was removed from the study. This is because males and females exhibit different disease and survival profiles.

However, some questions remain that what kinds of effect " females” have on Radium-224 exposure?

3. Since cancer and non-cancer are competing causes of death.

Can the authors do a Competing risk model in Statistical analysis?

Or cause-specific hazard function?

4. Can the authors address some limitation of this study at the section of Discussion.

Reviewer #2: [General comments]

This manuscript focuses on mortality from non-cancer diseases and all causes rather than specific types of cancer which may or may not be associated with therapeutic exposures to ionizing radiation among ankylosing spondylitis patients. The authors demonstrate no significant increase of overall mortality in relation to the Ra-224 treatment based on the analyses of the archived follow-up data. However, a caution should be paid in interpreting the results because the quality of follow-up were likely to be different for different populations (patients with or without Ra-224 treatment). In addition, baseline characteristics were likely to be different between patients with and without Ra-224 treatment. In these points, the authors should describe a little bit detail about how to follow the study subjects (with reference to the relevant previous study), and the reason why some of the patients were not treated with Ra-224. Without relevant explanations, the authors cannot come to the conclusion. The following are specific comments.

[Specific comments]

#1, 1st paragraph, page 4, METHODS

How to follow-up for mortality should be described at least briefly. In addition, the reviewer wonder if the authors used information on date of exit (date of last observation) for each study subject, but there was no description in the text.

#2, 2nd paragraph, page 5, METHODS

The authors described “The Cox model allowed for the different ages of the patients entering the study”, but it is unclear if they used attained age of study subjects as time scale in the Cox model, or they used age at entry as a stratified variable or a covariate in the Cox model. Please make it clear.

#3, line 1-2, page 6, RESULTS

The proportions of the deceased subjects were different for three populations. The possible reason for it and its potential impact should be carefully discussed in DISCUSSION section.

#4, line 4 and 11, page 6, RESULTS,

Table 1 was referred but it was missing in the paper.

#5, line 8, page 6, RESULTS,

Confidence interval was shown, but either 90% or 95% CI was unclear. Please specify it here and later on where appropriate.

#6, line 12, page 6, RESULTS,

“The odds ratio” was shown here, but it should be “hazard ratio.” In addition, RRs should be revised to HRs through the text.

#7, line 14, page 6, RESULTS,

There was a typo; RR of “098.” It should be revised.

#8, line 1, page 7, RESULTS,

Figure 2 was referred here, but Figures 3-6 were already referred earlier. Please reorder the figures.

#9, 1st sentence of 2nd paragraph, page 8, DISCUSSION,

The authors discussed about their assumption that radium-treated and control populations were identical in all respects except for dose from administered Ra. They should carefully explain the justification of their assumption.

#10, Fig. 2,

The proportions of selected causes of death were shown for three population, but a table which shows numbers of deaths and crude rates is more informative than this kind of figure.

6. PLOS authors have the option to publish the peer review history of their article (what does this mean?). If published, this will include your full peer review and any attached files.

Reviewer #1: No

Reviewer #2: No

---

## [Author Response · Author response to Decision Letter 0]

26 Feb 2020

Responses to the reviewer comments which were very helpful:

Reviewer 1:

#1: All the references have been changed to the Plos One style as well as the citations in the text.

#2: On line 88 we added the fact that the females consisted of less than 10% of the patients. Including them would increase the complexity of the analysis. 

#3: The analysis using Cox Proportional Hazard Regression does adjust for competing risks.

#4: Lines 221-223 were added to the discussion section which suggests an additional concern with our conclusions.

Reviewer 2:

Material has been added to the discussion. Also the non-treated patients were those attending a hospital that objected to using radium in treatment. A sentence was added to the discussion about this being a concern.

#1: Line 122 was added which explains the follow-up.

#2: Line 119 adds more detail about the variables in the Cox model.

#3: Line 419 begins an explanation for the differences in proportion of deaths (age).

#4: Table 1 is now included. This was a simple error by not having included it.

#5: All confidence intervals now clearly state that they are 95% intervals.

#6: HRs are now used through out the text.

#7: This is corrected. Line 153.

#8: This has been corrected.

#9: It is now stated that for the analysis it was necessary to assume that the population were identical. We continue to say they actually are not (different average ages) and different hospitals etc.

#10: I believe Fig. 6 is meant here and not Fig. 2. We prefer to present a figure instead of a table of number of cases especially when the various groups have different aver ages. If necessary we could simply remove the figure from the manuscript.

---

## [Decision Letter · Decision Letter 1]

20 Apr 2020

Administration of Lower Doses of Radium-224 to Ankylosing Spondylitis Patients Results in No Evidence of Significant Overall Determent

PONE-D-19-26797R1

Dear Dr. Hoel,

We are pleased to inform you that your manuscript has been judged scientifically suitable for publication and will be formally accepted for publication once it complies with all outstanding technical requirements.

With kind regards,

Suminori Akiba, M.D., Ph.D.

Academic Editor

PLOS ONE

Additional Editor Comments (optional):

Reviewers' comments:

Reviewer's Responses to Questions

**Comments to the Author**

1. If the authors have adequately addressed your comments raised in a previous round of review and you feel that this manuscript is now acceptable for publication, you may indicate that here to bypass the “Comments to the Author” section, enter your conflict of interest statement in the “Confidential to Editor” section, and submit your "Accept" recommendation.

Reviewer #2: All comments have been addressed

2. Is the manuscript technically sound, and do the data support the conclusions?

Reviewer #2: Yes

3. Has the statistical analysis been performed appropriately and rigorously? 

Reviewer #2: Yes

4. Have the authors made all data underlying the findings in their manuscript fully available?

Reviewer #2: Yes

5. Is the manuscript presented in an intelligible fashion and written in standard English?

Reviewer #2: Yes

6. Review Comments to the Author

Reviewer #2: All the comments from reviewers have been addressed and the manusccript seems to be revised satisfactory. However, there are several typographic errors including the followings, and I would like to suggest the authors to check English throuout the manuscript.

line 212 "RR" should be read as "HR"

line 220: "older then" should be read as "older than"

line 240: "The results this study" should be read as "The results of this study"

7. PLOS authors have the option to publish the peer review history of their article (what does this mean?). If published, this will include your full peer review and any attached files.

Reviewer #2: No

---

## [Editor Report · Acceptance letter]

22 Apr 2020

PONE-D-19-26797R1 

Administration of Lower Doses of Radium-224 to Ankylosing Spondylitis Patients Results in No Evidence of Significant Overall Determent 

Dear Dr. Hoel:

I am pleased to inform you that your manuscript has been deemed suitable for publication in PLOS ONE. Congratulations! Your manuscript is now with our production department. 

With kind regards,

on behalf of

Dr. Suminori Akiba 

Academic Editor

PLOS ONE